# Isolation and Characterization of *Beauveria caledonica* (Ascomycota: Hypocreales) Strains for Biological Control of *Odoiporus longicollis* Oliver (Coleoptera: Curculionidae)

**DOI:** 10.3390/microorganisms13040782

**Published:** 2025-03-28

**Authors:** Mingbi Ding, Li Wu, Hongwei Yu, Huacai Fan, Zhixiang Guo, Shengtao Xu, Jianhui Chun, Yongfen Wang, Si-Jun Zheng

**Affiliations:** 1Yunnan Key Laboratory of Green and Control of Agricultural Transboundary Pests, Agricultural Environment and Resources Institute, Yunnan Academy of Agricultural Sciences, Kunming 650205, China; dingmingbi@stu.ynu.edu.cn (M.D.); wucli9854@163.com (L.W.); hwei_yu@163.com (H.Y.); hcfan325@126.com (H.F.); zhixiangg@163.com (Z.G.); sj_xushengtao@163.com (S.X.); 2Resource Plant Research Institute, Yunnan University, Kunming 650205, China; chunjianhui@stu.ynu.edu.cn; 3Institute of Tropical and Subtropical Economic Crops, Yunnan Academy of Agricultural Sciences, Baoshan 678000, China; 4Bioversity International, Kunming 650205, China

**Keywords:** *O. longicollis* (Oliver), *B. caledonica*, biological characteristics, virulence, greenhouse evaluation

## Abstract

The banana pseudostem weevil (BPW), *Odoiporus longicollis* (Oliver), is one of the most destructive pests of bananas that is seriously affecting the yield and quality of bananas. We isolated pathogens from banana pseudostem weevils in Xishuangbanna and Dongchuan, Yunnan, China, and explored their biological characteristics. The pathogenicity of the strains was verified through laboratory and greenhouse inoculation experiments. The results showed that four strains of fungi were identified and confirmed as *Beauveria caledonica* (*Bc*) via ITS-rDNA sequencing. Optimal in vitro culture conditions were found to be a photoperiod of 24 h light, 25 °C temperature, and 18 days on potato dextrose agar (PDA) medium with insect meal. Under these conditions, the Cs-1 strain achieved a colony diameter of 65.17 ± 0.74 mm and spore production of 1.24 × 10^8^ cfu/cm^2^. The Cs-1 strain had the shortest lethal time (LT_50_) of 9.36 days at an inoculum of 1.00 × 10^9^ cfu/mL, with a lethality of 86.67% after 20 days. The Cs-3 strain showed 77.78% lethality at 1.00 × 10^8^ cfu/mL after 20 days. Despite variations in virulence, lethality did not correlate with major cuticle-degrading enzymes. The Cs-3 strain demonstrated effective biocontrol in greenhouse tests. Banana plants suffered significant damage without *Bc*-treated BPW, while the treated plantlets thrived. The mortality rate reached 82.78% after 35 days. This study marks the first identification of these entomopathogenic fungi (EPF) in Yunnan, China, highlighting *B. caledonica*’s potential for biocontrol application.

## 1. Introduction

Tropical and subtropical bananas and plantains (*Musa* spp.; ‘bananas’ hereafter)—the fourth largest food crop in developing countries—supply markets throughout the world [1]. The total global production reached 179 million tons in 2022, of which 44 million tons were plantains and cooking bananas [2]. China is one of the world’s largest producers and consumers of bananas. According to the Food and Agriculture Organization of the United Nations (FAO), by 2020, China’s net imports of bananas ranked third globally, accounting for 9 percent of the global imports [2]. As China’s banana industry improves in terms of cultivation techniques and pest control, overall banana production is increasing year by year, although the area under banana cultivation is decreasing [3].

However, alongside the urgent development of the banana industry, increasing biotic and abiotic stresses pose a significant threat to its sustainability [4]. Over 400 species of insects and mites attack bananas [5]. Among those insects, the banana pseudostem weevil (BPW) is one of the most damaging pests [6]. The BPW belongs to the Coleoptera order and the Curculionidae family of weevils. The BPW is slightly larger in size than *Cosmopolitus sordidus* (Germer), and its abdomen cannot be completely covered by the elytra. Some studies also suggest that its color is closely related to gender [7]. This pest feeds on the banana pseudostem during both its larval and adult stages, boring dense longitudinal and transverse holes that hinder the plant’s ability to transport nutrients [8,9,10]. The life cycle of the BPW consists of the four following developmental stages: eggs, grubs (from the first instar to the sixth instar), pupae, and adults. The incubation period of eggs is 3–5 days. The total survival period of larvae is 21–23 days. The pre-pupal survival period is 2–3 days, the survival period of the pupae is 8–10 days, and the survival period of adults is approximately 65–200 days [11]. In severe cases, this damage can lead to rot or even collapse of the banana plant, significantly impacting both yield and quality [12]. Reports indicate that BPWs can cause economic losses to the banana industry, ranging from 10% to 90% [12,13,14]. Furthermore, because the BPW primarily resides within the banana pseudostem, its presence is difficult to detect, and the actual incidence may be much higher than reported [15]. The BPW’s attraction to banana pseudostem volatiles [16], combined with the strong migratory capabilities of adult weevils [17], as well as climate change and other factors, have exacerbated damage in many major banana production areas [18]. Therefore, developing effective control measures for the BPW has become a critical issue for banana growers.

The current control measures for the BPW primarily rely on chemical methods [19]. However, since the BPW is predominantly active within the interior of the pseudostem, the insect’s exposure is significantly limited [20]. Pesticide applications also contribute to environmental pollution, with severe and destructive impacts on the ecosystem [21,22]. Because traditional trapping techniques are often time-consuming, labor-intensive, and ineffective [23], and cover crops generally prove ineffective [24], the adoption of green control measures offers potentially valuable alternatives to control BPWs.

Green control is a crucial measure for the biological control of agricultural pests and a key strategy for promoting pesticide reduction and management [25,26]. Consequently, green control has been of great interest in pest control research. Entomopathogenic fungi (EPF) is a fungus that uses insects as hosts and spreads widely within communities, causing disease and even death in the host insects [27]. Its high virulence, ease of transmission, resistance, and environmental compatibility as a bioinsecticide will add a new dimension to pest control [27,28]. More than 700 species of EPF have been identified [29]. Among these, the genus *Beauveria* has broad insecticidal properties and strong pathogenicity [30,31,32]. The primary species include *Beauveria bassiana*, *Beauveria brongniartii*, *Beauveria amorpha,* and *B. caledonica* (hereafter referred to as *Bc*) [33,34,35]. Among them, *B. bassiana* has been the most intensively researched, with numerous studies demonstrating its exceptional pathogenicity and efficacy in pest control [36,37]. Entrepreneurs have also commercialized *B. bassiana* [36], underscoring its significance in biological control.

*Beauveria caledonica* was first reported to have been isolated from Scottish bog soils and was not classified as an entomopathogenic fungal species at that time [38]. Huang et al. [39] isolated a strain of *Bc—*identified based on internal transcribed spacer (ITS) sequences and morphological characteristics—from the carcass of the root borer *Scolytidea*. Subsequently, Glare et al. [40] assessed the pathogenicity of the strain to the host, proving that it is a natural pathogen of *Hylastes ater* and *Hylurgus ligniperda* (*Curculionidae: Scolytinae*). A study by Mascarin et al. [41] found that *Bc* was more pathogenic than *B. bassiana* against *Cosmopolites sordidus*. Therefore, *Bc* has significant potential as a biocontrol agent for managing Coleoptera pests in agroforestry.

Our team discovered naturally occurring BPW adults and larvae affected by fungal disease in banana plantations during Fusarium wilt surveys conducted in two banana-producing regions in Yunnan, China. In this study, we aimed to isolate and identify *Bc* for the infected insects by morphological identification and molecular biology ITS-IDNA sequence analysis to study the biological characteristics of the isolated strains, including the optimum medium, light, temperature, and pH. We also further explored the pathogenicity of isolated strains against the BPW at various concentrations to assess its effectiveness for its potential biocontrol application under greenhouse conditions.

## 2. Materials and Methods

### 2.1. Bc, BPW, and Plant Material

Adults and larvae of diseased BPWs were collected from banana plants at the Banana Experimental Base of Yunnan Academy of Agricultural Sciences, located in Manjinlong Village, Gashan Town, Jinghong City, Xishuangbanna Prefecture, Yunnan Province (21°54′ N, 100°48′ E, 623.5 m above sea level), as well as at the banana plantation in Inmin Town, Dongchuan District, Kunming (26°19′ N, 102°55′ E, 1042 m above sea level). Diseased weevils collected in the field were individually stored in 5 mL centrifuge tubes and transported to the laboratory for isolation and identification. The diseased weevils were placed in a sterile Petri dish, and their surfaces were washed with 5 mL of sterile water. They were then transferred to 75% ethanol for 5 s, followed by three washes with sterile water. Afterward, the weevils were placed on sterilized filter paper in a Petri dish for 2 to 3 min. Once the surfaces of the diseased weevils were completely dried, the weevils were divided into three to five pieces using tweezers. These pieces were then inoculated into potato dextrose agar (PDA) medium and incubated at a constant temperature of 25 ± 1 °C for cultivation. The colonies that developed on the plates were numbered sequentially. After the colonies had grown, those exhibiting different morphologies were identified and numbered. The colonies were then extracted in order, according to their numbers, and inoculated onto new plates. The resulting isolates were further purified and stored at −80 °C.

All adult BPWs involved in this study were captured from banana plantations in Xishuangbanna Prefecture. Every 50 heads were packed into a sterile insect culture box (L: 25.8 cm, W: 18.2 cm, H: 11 cm). The colony fed on fresh and healthy banana pseudostem slices—the pseudostems of banana plants from banana plantations that were free of pests and diseases. The insects were isolated in the laboratory for 60 days under full darkness at a temperature of 25 °C and a relative humidity of 75%. During this time, fresh banana pseudostem slices were replenished, and the residual food waste was cleaned out of the box every 2 days. The status of the BPWs was observed every day. If dead insects were detected, they were taken out of the box immediately with sterilized tweezers and then placed in sterile Petri dishes. The insect carcasses were then examined for fungal infection, and the box was sterilized while replacing the pseudostem slices for treatment. Only insects without signs of infection were used in the experiment throughout the 60-day period.

In this study, the Cavendish banana cultivar “Baxi” (*Musa* spp. AAA) was utilized for the propagation of banana plantlets through in vitro propagation. The plantlets were incubated at 25 °C and L//D = 16 h//8 h until new roots grew and were then transplanted into hole trays of size 54 × 28 cm (32 holes). When the banana plantlets grew to a plant height of 15 cm and 5 leaves, they were transplanted into plastic pots with a height of 17.5 cm and diameter of 20.7 cm (garden soil: substrate = 1:1). All banana plantlets were grown in a solarium.

### 2.2. Strain Culture and Morphological Observation

The strains obtained from isolation and purification were inoculated onto the PDA medium using an inoculating needle in an ultra-clean bench and placed in an incubator at 25 ± 1 °C. After 14 days and 30 days, the strains were removed to observe the morphological characteristics of the colonies.

After scraping a small amount of mycelium (0.5 × 0.5 × 0.1 mm^3^) from the plate with a sharp blade (during this process, care was taken to avoid pulling, contusing, and squeezing), the sample was immediately placed in a 2.5% glutaraldehyde solution for fixation for 24 h. It was then rinsed three times with phosphate-buffered saline (PBS) for 15 min each time. After that, the sample was subjected to dehydration treatment, critical-point drying, and gold coating. Subsequently, the mycelial morphology, spore-producing structures, and conidial morphology of the colonies were observed under a scanning electron microscope (Hitachi Reguls 8100, Tokyo, Japan). The samples were photographed and recorded and then classified and identified according to the morphology of the strains.

### 2.3. Molecular Biology Identification

We extracted fungal DNA by referring to the Omega D3390 kit (Omega D3390, Tsingke, Fuzhou, China). The strain genomic DNA was used as the template for PCR amplification of the strain rDNA-ITS sequence using the fungal universal primers ITS1 (5′-TCCGTAGGTGAACCTGCGG-3′) and ITS4 (5′-TCCTCCGCTTATTGATATGC-3′) [42]. The PCR reaction system was 50 μL, including 2 μL (400 nM) of each of the upstream and downstream primers ITS1 and ITS4, 45 μL of the PCR MasterMix (2× FidCycle Fast, Sangon Biotech, Shanghai, China), and 1 μL of the template DNA (180 ng). The PCR reaction conditions were as follows: pre-denaturation at 98 °C for 2 min, denaturation at 98 °C for 10 s, annealing at 72 °C for 10 s, and extension at 72 °C for 5 min for 35 cycles. The PCR-amplified products were detected by 1% agarose gel electrophoresis and sent to Beijing Prime Biotechnology Co. (Beijing, China). The sequencing results were compared with the National Center for Biotechnology Information (NCBI) database to obtain the corresponding sequences of multiple strains with homology to the target strains, from which the sequences with higher homology were downloaded. The phylogenetic tree of the strains was constructed and analyzed by using the neighbor-joining method (NJ) with 1000 bootstrap validations using MEGA 7.0 software. The phylogenetic tree of the isolated strains was constructed and analyzed.

### 2.4. Biological Characterization of Isolated Strains

#### 2.4.1. Effect of Temperature on the Growth and Spore Production of Strains

The conidia of the test strain were prepared as a spore suspension at a concentration of 1.0 × 10^8^ cfu/mL using 0.05% Tween-80, following the bacterial droplet method [43]. The bacterial solution of 2 μL was aspirated with a pipette gun and inoculated into the center of the plate of the PDA medium. The Petri dishes were then incubated at four different temperatures—20 °C, 25 °C, 28 °C and 30 °C—with five replicates for each treatment. At 18 d, the diameter of the colonies was measured by the “crosshatch method” with a vernier caliper (0.01 mm) [44]. The bacterial discs were punched at a 1/2 radius of each colony with a 0.5 cm diameter hole puncher. Then, they were placed in 50 mL conical flasks. The flasks, with 20 mL of 0.05% Tween-80 sterile solution added, were placed on a magnetic stirrer and stirred thoroughly. The spore concentration was measured by a hemocyte counting plate (0.10 mm, 1/400 mm^2^) and converted to conidial production per unit area to determine the optimal culture temperature.

#### 2.4.2. Effects of Photoperiod on Growth and Spore Production

The four strains were inoculated in the center of the PDA medium using the method described above (Section 2.4.1). Then, the medium was incubated in five light incubators with photoperiods of L//D = 12 h//12 h, L//D = 24 h//0 h, L//D = 0 h//24 h, L//D = 16 h//8 h, and L//D = 8 h//16 h. The incubation was repeated in five plates for each treatment. The diameter and conidial yield were measured at 18 d.

#### 2.4.3. Influence of the Medium on Growth and Spore Production

The four strains were inoculated at the center of the PDA, sorghum dextrose agar yeast extract (SDAY), iPDA (i.e., PDA + 0.5% insect powder), and iSDAY (i.e., SDAY + 0.5% insect powder) medium plates using the method of Section 2.4.1 [45]. The BPW yielded insect powder by drying it in an oven at 104 °C for 8 h, grinding it into a fine powder using a mortar and pestle, sifting it through an 8-mesh sieve, collecting it, and setting it aside. The media were then incubated under optimal temperature and light conditions, respectively. Each treatment was repeated in five plates. The colony diameter and conidial production were measured after 18 d.

#### 2.4.4. Effect of pH on Growth and Spore Production

The four strains were inoculated at the center of the PDA medium with pH levels of 6, 7, 8, 9, 10, and 11 using the method described above in Section 2.4.1. The medium was then incubated under optimal temperature and light conditions. Each treatment was replicated across five plates. The colony diameter and conidial production were measured after 18 d.

### 2.5. Enzyme Activity Assay

After cultivating *Bc* in the PDA medium for 15 d, the biomass was collected, and moisture was removed. A 60 mg sample was weighed, ground in liquid nitrogen, and resuspended in PBS at a ratio of 1:9. The sample was sonicated at 300 W for 5 s, with 5 s intervals, repeated five times. After sonication, the mixture was vortexed and shaken and then centrifuged at 5000 r/min for 15 min to collect the supernatant for further analysis.

The enzyme activities of the four strains of *Bc* related to cuticle degradation were determined according to the manufacturer’s instructions using the Microbial Protease, Chitinase, and Lipase Activity Assay Kit (Tsingke Biotechnology Co., Ltd., Beijing, China).

### 2.6. Pathogenicity Determination

#### 2.6.1. Preparation of Spore Suspensions

Each strain was inoculated onto a PDA medium and cultured at a temperature of 25 ± 1 °C and a photoperiod of L//D = 12 h//12 h for 15 d. After the strain was cultured and spores were produced, the conidia on the surface of the colonies were scraped with a sterile blade into a triangular vial of 0.04% Tween-80 sterile aqueous solution. Mycelium was filtered through a double-layer of sterile gauze and stirred sufficiently in a magnetic stirrer to make the spores uniformly dispersed in the solution. The spore concentrations were determined by a hemocytometer plate and deployed as 1.00 × 10^9^ cfu/mL, 1.00 × 10^7^ cfu/mL, and 1.00 × 10^5^ cfu/mL.

#### 2.6.2. Pathogenicity Test

Healthy adults, after 60 d of indoor rearing, were rinsed three times in sterile water to remove plant residues and were then immersed in different concentrations of conidial suspensions for 10 s. The control colonies were immersed in sterile water containing 0.04% Tween-80. The swarms were then individually placed in sterilized culture boxes (L: 25.8 cm, W: 18.2 cm, H: 11 cm) with slices of banana pseudostem as a food source. They were reared under dark conditions in the growth chamber room at a temperature of 25 °C and 75% relative humidity for 25 d. Three replicates were included for each treatment, with 15 individuals in each replicate. The mortality rate was recorded daily. All deceased insects were surface disinfected with 1% sodium hypochlorite and 70% ethanol, then rinsed in sterilized water three times and placed in sterile Petri dishes. In order to determine whether the deaths were caused by *Bc*, the growth of mycelium or conidia on the surface of the insects was observed at regular intervals to examine their morphology microscopically.

#### 2.6.3. Comparative Test of Pathogenicity of Different Strains

Using the test method described above in Section 2.6.2, four strains of BPW were treated with spore suspensions of 1.00 × 10^8^ cfu/mL to investigate the degree of pathogenicity of each strain of *Bc*, respectively.

### 2.7. Evaluation of the Control of BPW by B. caledonica in Greenhouse Potted Plantlets

*Bc* has high pathogenicity against BPWs in the laboratory. To validate the laboratory results, the present study added a greenhouse pot experiment to evaluate the biocontrol efficacy of *Bc* against BPWs, utilizing banana plants that were one year old as the host. The plants infested with BPWs were treated with *Bc*. Two treatments were established: (1) the control (CK group), where BPW adults were introduced into banana plantlets without prior inoculation of *Bc*, and (2) the treatment (Cs-3 group), where BPW adults inoculated with *Bc* in advance were introduced into banana plantlets. The inoculum concentration of *Bc* was 1.00 × 10^8^ cfu/mL, while sterile water was used to replace the spore suspension in the control group. Each treatment was replicated five times, with eight banana plants placed in each insecticidal cage (L: 100 cm; W: 100 cm; H: 100 cm) as one replicate. A total of 60 adult BPWs were introduced into each cage, totaling 80 banana plants and 600 BPWs. The specific inoculation method was the same as that described in Section 2.6.2 above; all the treated banana plantlets were cultured and monitored in the greenhouse of the Yunnan Academy of Agricultural Sciences.

The growth of the banana plantlets was observed and recorded at 35 d after inoculation. The plant height, stem thickness, number of leaves, and fresh weight of banana plantlets were measured from different treatments. Photographs were taken of each plantlet. Symptoms of BPW, symptoms of BPW onset, and the incidence rate of BPWs across different treatments were also recorded.

### 2.8. Data Analysis

Duncan’s test for the significance of differences in the colony diameter and hug production per unit area was conducted using SPSS-22.0. The Probit method was used to find the regression equations and to calculate the lethal mid-concentration (LC_50_) and lethal mid-time (LT_50_). The graphs were plotted using GraphPad Prism 8.

For this study:Colony diameter (mm) = (colony transverse diameter + colony longitudinal diameter)/2;(1)Spore production per unit area of the colony (spores/cm^2^) = number of spores per cell × 400 × 10^4^ × dilution/(3.14 × r × *n*), with r being the radius of the colony (cm) and n being the number of colonies (nuggets);(2)Cumulative mortality rate (%) = (total cumulative number of the insects killed in the treatment group/total number of the insects treated) × 100;(3)Corrected cumulative mortality rate (%) = (cumulative mortality rate in the treatment group − Cumulative mortality rate in the control group)/(1 − control mortality rate) × 100.(4)

## 3. Results

### 3.1. Strain Morphological Characterization

#### 3.1.1. Field Characteristics

Adults and pupaes of the infected BPWs were observed in the outer layer of the banana pseudostem. At the onset of the disease, a dense white mycelium adhered to the body segments and rostrum of the insect, proliferating and eventually enveloping it, which resulted in the production of a substantial quantity of white powdery spores (Figure 1).

#### 3.1.2. Micromorphological Characteristics

The pathogenic fungus was isolated and purified, after which the strain was inoculated onto a PDA medium. Initially, the colonies appeared white and round, but as spore production occurred, the surface changed to a cream color. The spore layer had a powdery texture. The abaxial surface of the colony displayed a yellowish hue (Figure 2A–D; Appendix A). Under the scanning electron microscope, the mycelium was colorless and smooth, with a diameter of 1.5–2.0 μm. The sporulating cells were densely clustered on the mycelium. Conidial peduncle or expanded vesicles, spherical to bottle-shaped, with the size of sporulating cells, ranged from 2.4–5.7 μm × 1.9–3.3 μm. The sporulating axes were about 14 μm in length, with a small denticle on the axes, curved in the form of a knee. The spore-producing cells and vesicles often multiplied, aggregating into dense spore heads on the conidial peduncles or hyphae. The conidia were colorless, smooth, unicellular, mostly short cylindrical, rarely ellipsoidal, and usually bluntly rounded at both ends. The conidium size was 2.5–6.5 μm × 1.3–2.2 μm (Figure 2E–G).

### 3.2. Sequence Analysis of Strain rDNA-ITS

The fragments of the PCR amplification products of the rDNA-ITS from the four strains were determined. The sequences were 196 bp. The phylogenetic tree was constructed using the neighbor-joining method with the *Methanobacterium alcaliphilum* strain NBRC 105226 NR_112910.1:1-453 as the reference bacterium. Figure 3 shows the evolutionary tree. The comparison results showed that the isolates were 99.35%, 99.36%, 99.45%, and 99.78%, similar to *Bc* BCRC 32867 NR 077147.1 (accession no. NR 077147.1), respectively. Consequently, the combined morphological analysis and rDNA-ITS sequence similarity analysis identified the strain as *B. caledonica*.

### 3.3. Effect of Different Culture Conditions on the Growth of Strains

The effect of different light conditions on the growth of the strains showed that the four *Bc* strains could grow under five light conditions: L//D = 12 h//12 h, L//D = 24 h//0 h, L//D = 0 h//24 h, L//D = 16 h//8 h, and L//D = 8 h//16 h. The colony diameter was measured at 18 d. The results showed that Cs-1, Cs-2, and Cs-3 had the largest colony diameters under the photoperiod of L//D = 24 h//0 h, with diameters of 54.03 ± 0.65 mm, 50.94 ± 3.32 mm, and 50.94 ± 3.32 mm, respectively, which were significantly higher than that under the photoperiod of L//D = 0 h//24 h. The diameters of Cs-5 were significantly higher than that under L//D = 8 h//16 h. The diameters of Cs-5 were significantly higher than those under L//D = 8 h//16 h. The diameter of Cs-5 was the largest under the light condition of L//D = 8 h//16 h, which was 48.67 ± 1.26 mm. However, it was not significantly different from that under the condition of L//D = 24 h//0 h. In conclusion, the diameters of the four strains cultivated for 18 days under the treatment of L//D = 0 h//24 h were significantly lower than the colony diameters under L//D = 12 h//12 h, L//D = 24 h//0 h, L//D = 16 h//8 h, and L//D = 8 h//16 h. Supplemental light can promote the growth and sporulation of *Bc* (Figure 4A).

Spore dispersal is an important route for the spread of pathogenic fungi. Therefore, this study investigated the spore production of *Bc* under different culture conditions to determine the optimal conditions. We found that the production of the Cs-1 strain with L//D = 24 h//0 h and L//D = 12 h//12 h treatments were 1.35 × 10^8^ cfu/cm^2^ and 1.29 × 10^8^ spores/cm^2^, respectively, which were significantly higher than the other treatments. Spores/cm^2^ for the Cs-1 strain were also significantly higher than the other treatments. For the Cs-2 strain, the L//D = 24 h//0 h and L//D = 16 h//8 h treatments produced 1.07 × 10^8^ cfu/cm^2^ and 9.64 × 10^7^ cfu/cm^2^, which were significantly higher than the other treatments, respectively. For the Cs-3 strain, the L//D = 24 h//0 h and L//D = 8 h//16 h treatments produced 4.7 × 10^8^ cfu/cm^2^ and 9.64 × 10^7^ cfu/cm^2^, respectively. The spore production of the Cs-3 strain L//D = 24 h//0 h and L//D = 8 h//16 h treatments were 4.16 × 10^7^ cfu/cm^2^ and 4.50 × 10^7^ cfu/cm^2^ respectively, which were significantly higher than the other treatments. The spore production of the Cs-5 strain L//D = 8 h//16 h treatment for 18 d was 1.18 × 10^8^ cfu/cm^2^, which was significantly higher than the other treatments, followed by the L//D = 24 h//0 h treatment (Figure 4B).

The morphology of the colonies was observed after 18 days. The colonies were round and white, with a slightly pink center and a dry surface featuring ring-like protrusions. In conclusion, full light conditions are most suitable for the spore production of *B. caledonica*, while dark conditions are unfavorable for spore formation (Figure 4C–G).

By setting different incubation temperatures, we conclude that *Bc* could grow at 20 °C, 25 °C, and 28 °C. However, all four strains ceased to grow when the temperature rose to 30 °C. At 28 °C, while all strains exhibited some growth, the colony diameters were significantly lower than those at 20 °C and 25 °C. After 18 days of incubation, the colony diameters of the four strains reached 64.25 ± 1.40, 57.41 ± 1.69, 55.33 ± 1.12 mm, and 63.05 ± 1.63 mm colony diameters at 25 °C, which were significantly higher than the diameters at the remaining three temperatures (Figure 5A).

Under different temperature conditions, the spore production of *Bc* at 25 °C was significantly higher than that of other treatments. After 18 days, the spore production per unit area for Cs-1, Cs-2, Cs-3, and Cs-5 was 1.21 × 10^8^, 8.20 × 10^7^, 1.02 × 10^8^ cfu/cm^2^ and 9.85 × 10^7^ cfu/cm^2^, respectively (Figure 5B).

After 18 d at 20 °C, the colonies appeared with a white color and a slightly yellow center, with ring-like protrusions. The surface was moist with water droplet formations. At 25 °C, the colonies were circular, and the surface was cream-colored. The spore layer was powdery. Water droplets formed on the wet surface. In conclusion, the optimal growth temperature of *Bc* is 25 °C. Culture temperatures that are too high will lead to the cessation of growth or even death of the strain (Figure 5C–F).

The effect of the culture medium on the growth of *Bc* indicated that the PDA medium was more conducive to the growth of *Bc* compared to the SDAY medium. Simultaneously, the addition of insect powder enhanced the growth of *Bc* in both mediums. By 18 d, the diameter of the colonies under the iPDA medium conditions had the largest colony diameters. The diameters of the four strains reached 65.17 ± 0.74, 60.21 ± 3.26, 56.99 ± 0.46, and 63.17 ± 0.7 mm, respectively. Among the four strains, there was no significant difference in the colony diameters of Cs-1, Cs-2, and Cs-5 in the iPDA and PDA media. There were no significant differences in the colony diameter between the iPDA and PDA media for Cs-1 and Cs-5. Colony diameters for the iPDA media were significantly higher than the PDA media for the Cs-3 strain. The iPDA and PDA media had significantly higher colony diameters at 18 d than the other two media for all four strains (Figure 6A).

The spore production per unit area of the four *Bc* strains on the iPDA medium was significantly higher than that of the other treatments. The spore production per unit area at 18 days was 1.01 × 10^8^, 1.24 × 10^8^, 9.34 × 10^7^, and 1.24 × 10^8^ cfu/cm^2^, respectively (Figure 6B).

After 18 d, the colonies in the PDA medium were round, with a cream-colored surface and a powdery spore layer. In the medium added with insect powder, the strain accelerated spore production. The spore layer cracked in the center. The surface was dry relative to that of the colonies in the PDA medium. In the SDAY and iPDA media, the colonies were dry, round, and white, with spherical protrusions in the center. In summary, the PDA medium was more suitable for the growth and spore production of *B. caledonica*. At the same time, the appropriate addition of BPW insect powder was beneficial to the growth and spore production of the strain (Figure 6C–F).

Under different pH conditions, at 18 d, the colony diameters of Cs-1 in the pH 11 treatment were significantly lower than the others. The colony diameters at pH 10 were significantly lower than the colony diameters at pH 8. However, there was no significant difference between the other treatments. The colony diameters of Cs-2 were significantly higher than those of the other treatments at pH 7 and 9. But, there was no significant difference between the two treatments. The colony diameters at pH 6 were significantly lower than those of the other treatments in Cs-3. The colony diameters of Cs-5 were the lowest at pH = 11. Cs-3 had a significantly lower colony diameter than the other treatments at pH 6. Cs-5 had the lowest colony diameter at pH = 11. However, combining the colony diameters of the four strains at 18 d, the differences were small and lacked regularity (Figure 7A).

After 18 days, at pH = 9, the spore production per unit area was significantly higher than the other treatments, except for Cs-2, in which the difference in spore production at pH = 7 and pH = 9 was not significant. The spore production was 6.24 × 10^7^, 7.73 × 10^7^, 6.50 × 10^7^, and 5.65 × 10^7^ cfu/cm^2^, respectively, for the other three strains at pH = 9 (Figure 7B).

After 18 d, the colonies were round and white, with ring-like protrusions and powdery spore rings in the center of the colony. Water droplets formed to create a wet surface. In summary, there was a lack of regularity in the growth of the strains under different pH conditions. The four strains of *Bc* were insensitive to changes in pH (Figure 7C–H).

Protease, chitinase, and lipase were found in the four strains of *Bc*. Cs-3 had the highest protease content of 556.99 ± 1.09 IU/L, which was significantly higher than the remaining three strains. Chitinase was lower among the three enzymes measured. Among the four strains, Cs-2 had the highest content of chitinase, followed by Cs-5, which had 228.22 ± 2.29 IU/L and 220.50 ± 0.12 IU/L, respectively. The values for Cs-2 and Cs-5 were significantly higher than those of the other two strains. The lipase content was still the highest in Cs-2 and significantly higher than that of the other three strains, with a content of 480.60 ± 0.60 IU/L, followed by the Cs-5 strain. The content of chitinase was basically the same as that of lipase, whereas there was an opposite trend for the protease content (Appendix A).

### 3.4. Pathogenicity of B. caledonica Against O. longicollis (Oliver)

Regarding the influence of different inoculation concentrations of the Cs-1 strain on pathogenicity, the results showed that the cumulative mortality of BPWs increased with an increasing spore concentration of *Bc* and treatment time. The highest values of cumulative mortality were reached 20 d after inoculation at concentrations of 1.00 × 10^9^, 1.00 × 10^7^, and 1.00× 10^5^ cfu/mL. The highest cumulative mortality of BPWs was recorded at the highest concentration (1.00 × 10^9^ cfu/mL) treatment, with a mortality rate of 86 67% (Figure 8).

A time–dose effect characterized the pathogenic activity of *Bc* against the BPW. The LT_50_ shortened as the concentration of spore suspension increased. When the concentration was increased from 1.00 × 10^7^ to 1. 00 × 10^9^ cfu/mL, the larval LT_50_ decreased from 17.00 d to 9.36 d. However, the cumulative mortality of the BPW was below 50% when the spore concentration was 1.00 × 10^5^ cfu/mL, making it impossible to calculate the LT_50_ values (Table 1).

Applying the Probit model, the regression equation of pathogenic activity of *Bc* against the BPW on day 20 was obtained, where PROBIT(*p*) = −3.322 + 0.486x. The LC_50_ at day 20 was 6.898 × 10^6^ cfu/mL with a 95% confidence interval of 1.503 × 10^6^–2.967 × 10^7^ cfu/mL.

Pathogenicity is the most crucial index to measure the effectiveness of biocontrol. By testing the pathogenicity of the four test strains, the results showed that the cumulative mortality rate of the BPW after 20 d of inoculation with *Bc* for Cs-1, Cs-2, Cs-3, and Cs-5 were 63.89 ± 4.811%, 75 ± 8.333%, 77.78 ± 12.729%, and 63.89 ± 4.811%, respectively (Figure 9A). The magnitude of the pathogenicity of the four strains was determined based on the growth of fungi on the surface of the insects. The pathogenicity of the Cs-3 strain—77.78 ± 12.729% at 20 d—was significantly higher than that of the other three strains (Figure 9B). Enzymes help pathogenic fungi to break through the insect cuticle. In order to initially investigate the relationship between the pathogenicity and enzymes of *Bc*, a correlation analysis of pathogenicity and enzyme activity was carried out. There was a moderate positive correlation between protease activity and pathogenicity, but it was not significant (*r* = 0.478, *p* = 0.552) (Figure 9C). There was a moderate negative correlation between gibberellin activity and pathogenicity, but it was also not significant (*r* = −0.360, *p* = 0.640) (Figure 9D). Lipase activity had a weak negative correlation with pathogenicity, which remained insignificant (*r* = −0.115, *p* = 0.885) (Figure 9E). Therefore, there was no correlation between the pathogenicity and enzyme activity of the four test strains. Further studies are needed to screen high-pathogenicity strains from the perspective of enzyme activity.

### 3.5. Effectiveness of Greenhouse Control of B. caledonica

Researchers must determine the pathogenicity to successfully evaluate EPF. The complexity of the pest environments mean laboratory analysis is insufficient. Therefore, after determining the effectiveness of *Bc* against the BPW in the laboratory, we evaluated the interactions of the fungus and the pest on banana plants in the greenhouse. The results demonstrated that *Bc* was extremely effective against BPWs in potted plantlets.

After 35 d of treatment, the BPW in the control group caused serious damage to banana plants. All replicated banana plants suffered from different degrees of damage. The BPW was detrimental to the bulb, pseudostem, and the base of the leaves of the banana plantlets. BPWs fed on the banana plants, forming dense holes, which seriously affected the growth of the plants. Many of the plants even collapsed and rotted, making it impossible to grow. However, after the introduction of *Bc*-treated BPWs onto banana plants, the damage was significantly reduced. The plants grew well. The BPWs were fatally threatened (Figure 10).

BPWs seriously impacted the biological yield and physiology of banana plants. At 35 days, the height of the banana plants in the treatment group was significantly higher than that of the control group, reaching 45.14 ± 0.82 cm, compared to the control group’s 26.54 ± 2.77 cm (Figure 11A). At 35 d, the stem thickness of the banana plants in the control group reached 35.08 ± 1.67 mm, significantly lower than the 23.22 ± 1.27 mm of the treatment group (Figure 11B). After 35 d, the 7.31 ± 0.37 leaves per plant in the treatment group was significantly higher than the 3.29 ± 0.96 leaves per plant in the control group (Figure 11C). Finally, after 35 d, the fresh weight of the plants in the treated group reached 395.10 ± 30.69 g/plant, significantly higher than the 256.12 ± 29.99 g/plant found in the control group (Figure 11D).

In addition to determining the agronomic parameters of banana plants, we further evaluated the pathogenicity of *Bc* against BPWs in the greenhouse. The results showed that *Bc* was also highly effective against BPWs in the greenhouse environment, with a virulence of 82.93% after 35 d. The results showed an extremely strong pathogenic effect and suggested a high potential for using *Bc* to control BPWs (Figure 11E).

## 4. Discussion

The BPW poses a serious danger to the global banana industry, causing significant economic losses. Traditional control methods are inadequate for sustainable banana production. Therefore, there is a pressing need to develop effective and environmentally friendly biological pest management strategies. EPF offers a promising alternative to chemical pesticides [46] and could provide viable solutions for controlling BPWs. In this study, for the first time in China, we isolated the EPF *Bc* from diseased adults and nymphs of BPWs. *Bc* showed good pathogenicity against BPWs. *Bc* is a natural pathogenic fungus of BPWs, with lethality up to 86.67% when the concentration reaches 1.00 × 10^9^ cfu/mL (Table 1). The findings indicate that *Bc* shows strong pathogenicity towards BPWs, making it a valuable addition to the arsenal of EPF and providing a new tool for banana pest management.

Relatively little research has been conducted on *Bc*, which was initially isolated from Scottish soils and considered a non-insect species [38]. Subsequent research confirmed the action of *Bc* on insect pests such as *Hylastes ater*, *Hylurgus ligniperda*, and others [40]. Recently, *Bc* was found in plantain plantations in southeastern Baxi, infecting adults of *Cosmopolites sordidus*, where the isolate was even more pathogenic than *B. bassiana* [41]. At the same time, we also found an infestation of BPWs in the main banana growing area in Yunnan, China. These initial findings on the biological characteristics of *Bc* and its pathogenicity to BPWs require further systematic investigation, particularly under field conditions.

The biology of *Bc* has largely gone unstudied, leaving a knowledge gap and requiring future research. In this study, we tested the biology of four strains of *Bc* that were isolated from two banana plantations under different ecological conditions. The results showed that the temperature had the greatest effect on the growth of the four strains, with growth being severely impeded when the temperature was 28 °C. All four strains stopped growing when the temperature rose to 30 °C, indicating an important threshold for further growth of *Bc* strains. Since *Bc* has been less studied in pest control and nothing has been reported on BPWs, there is a lack of reference on the biology of *Bc*. However, Fargues et al. [47] studied a similar EPF, *B. bassiana*, from different geographic climates and host sources. They found that *B. bassiana* grew at temperatures ranging from 28 °C to 35 °C. The maximum temperature tolerance of all 65 *B. bassiana* strains exceeded 30 °C, suggesting that it is better adapted to high temperatures. This could explain why *Bc* and BPWs primarily reside in the banana pseudostem or the soil and are most active at night [48]. They avoid extreme heat. Clearly, this environmental factor must be considered in future practical applications. In addition to this, some studies have found that EPF could manage the host to find a suitable environment in the early and middle stages of infestation, which is necessary to prepare for the massive infestation in the later stages [49].

The pathogenicity of *Bc* against the BPW concerns us because it is the most important indicator of a pathogenic fungus’s potential for defense. The pathogenicity of EPF on target insects is by no means an accidental occurrence. It fully depends on the number of spores attached to the insect cuticle and does not appear to be dependent on whether or not the fungus is taken directly from the mouthparts [50]. This relies heavily on a set of infestation mechanisms developed by EPF that target the hard cuticle [51], allowing EPF to successfully reach the hemocoel and disrupt the insect defense system [52,53], as well as being more conducive to pathogenicity.

In this study, we infested healthy BPW adults with spore suspensions of *Bc* at different concentrations. We found that the cumulative mortality rate was positively correlated with the spore concentration. The time to lethality (LT_50_) was reduced when the concentration reached 1.00 × 10^9^ cfu/mL. The lethal median time to death (LT_50_) was 9.360 d at a concentration of 1.00 × 10^9^ cfu/mL (Table 1). This LT_50_ result was similar to Kisaakye et al. [54] for *Cosmopolites sordidus* using *B. bassiana* as the causal organism (5.3–11.1 d). The LC_50_ was 6.898 × 10^6^ cfu/mL by day 20, with a cumulative mortality rate of 86.67%. The LC_50_ was higher than that of Alagesan et al. [55] against BPWs, with *B. bassiana* as the causal agent at 27 d. The maximum mortality rate of 76.66% is consistent with the finding that *Bc* is more virulent than *B. bassiana*, as reported by Mascarin et al. [41]. Although there are some variations among strains, this result also confirms that *Bc* has good potential for pest control.

Our results showed that there were significant differences in pathogenicity among the different strains. The Cs-3 strain had the highest pathogenicity of 77.78% after 20 d when *Bc* was inoculated at a concentration of 1.0 × 10^8^ cfu/mL, showing a higher pathogenicity than the other three strains (Figure 9). We also determined the protease, chitinase, and lipase activities of the four strains, finding that all four strains contained high enzyme activities. These enzymes are extremely important substances in the process of breaking through the insect cuticle to enter the insect blood cavity when the entomopathogenic fungus infests the host insect [51,56,57,58]. We also analyzed the content and pathogenicity of the three enzymes in conjunction with each other. Our intent was to explore whether there is a correlation between pathogenicity and the three important enzymes. We found that the enzymes were related to pathogenicity and are a key factor in screening for highly pathogenic strains. However, the results showed no significant correlation between the three enzymes’ content and their virulence. The link between enzymes and virulence was also found to be inconclusive in previous studies on other strains. This finding coincides with many studies that did not yield reliable results [59,60,61]. Enzyme activity was not a significant factor in the screening of strains with high virulence. Thus, enzyme activity alone is not a significant criterion for selecting highly virulent strains, although it remains a key factor influencing virulence [62,63,64].

Future bioprophylactic evaluations for EPF may involve the stability of pathogenicity under complex environmental conditions. Most of the current tests of pathogenicity are based on laboratory conditions [55,65,66] despite the extremely complex environments that pests and their pathogenic bacteria live in. Therefore, we designed a mutualistic experiment involving banana plants, BPWs, and *Bc* in greenhouse pots to evaluate the efficacy of *Bc* against BPWs in relatively complex environments. The results showed that *Bc* had a very strong biocontrol capability. In the control group without *Bc* treatment, banana plants suffered serious damage from BPWs, which seriously affected the normal growth of the plants and even killed them. However, after applying *Bc* to BPWs in advance, in the treatment group, the BPWs died in large numbers, and the banana plants grew well. The virulence of *Bc* did not decrease under greenhouse conditions but rather increased, reaching 82.93% after 35 d (Figure 10E). Although there were differences between the two experiments conducted under different conditions, both strongly demonstrated the potential of *Bc* against BPWs. This greenhouse adaptability experiment showed that an EPF could be a great advantage for biological control [67,68]. It is highly competitive with traditional strain pathogenicity [69], especially in BPW control. Compared to *B. bassiana*, which has a large number of host insects and occurs in a wide range of areas, *Bc*’s use may be limited. However, its relatively single-host spectrum could be one of its advantages. For example, it could form an indirect protective effect on some non-target organisms such as silkworms, earthworms, and bees [34,70].

The way the BPW causes damage leads to its activity area being centered around individual banana plants. They are often concentrated in a single banana plant, and *Bc* spreads easily [25]. Some potential mechanisms by which insects might infect each other include secondary transmission through physical contact, shared food sources, etc. However, there is a lack of direct evidence in this regard in our current study. Therefore, it will be an interesting direction for future research in real-world environments. This greatly increases the likelihood of *Bc* spreading among the insect population and enhances its biocontrol potential. Therefore, further studies on *Bc* as a potential biocontrol agent should be conducted under field conditions in the next phase of research. As Mascarin et al. [41] found, *Bc* has antifungal activity against *Fusarium oxysporum* f. sp. *cubense.* We will also investigate its potential application in the management of Fusarium wilt of bananas in the next step.

## 5. Conclusions

In this study, the first of its kind in Yunnan, China, *Bc* was isolated from BPWs in two banana plantations. This study provides some baseline results on the biological characteristics of *Bc*. We demonstrated the potential of *Bc* in BPWs in terms of its pathogenicity and its potential to control BPWs. This is the first time that *Bc*, BPWs, and banana plants were studied in a greenhouse environment, demonstrating their strong pathogenicity and environmental adaptability. The results call for more research on the biological control of BPWs.

## Figures and Tables

**Figure 1 microorganisms-13-00782-f001:**
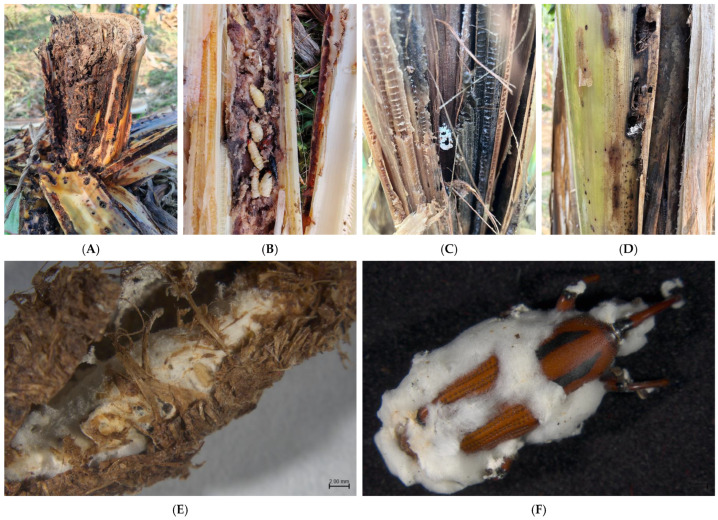
Field damage and onset symptoms of *O. longicollis* (Oliver): (**A**–**C**) field symptoms of *O. longicollis* (Oliver); (**D**): field-onset adult; (**E**): onset pupae; (**F**): onset adult.

**Figure 2 microorganisms-13-00782-f002:**
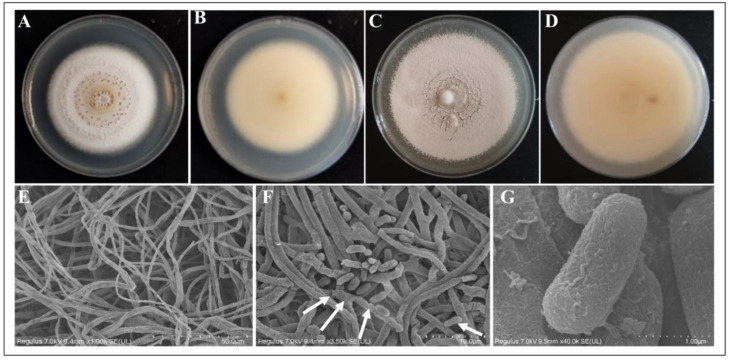
Micromorphological features of *B. caledonic*: (**A**): 14 d colony frontal surface; (**B**): 14 d colony abaxial surface; (**C**): 30 d colony frontal surface; (**D**): 30 d colony abaxial surface; (**E**): hyphae; (**F**): hyphae and conidia, with the arrows showing the peduncle, vesicle, sporulation cells, and conidia from left to right; (**G**): conidiophores.

**Figure 3 microorganisms-13-00782-f003:**
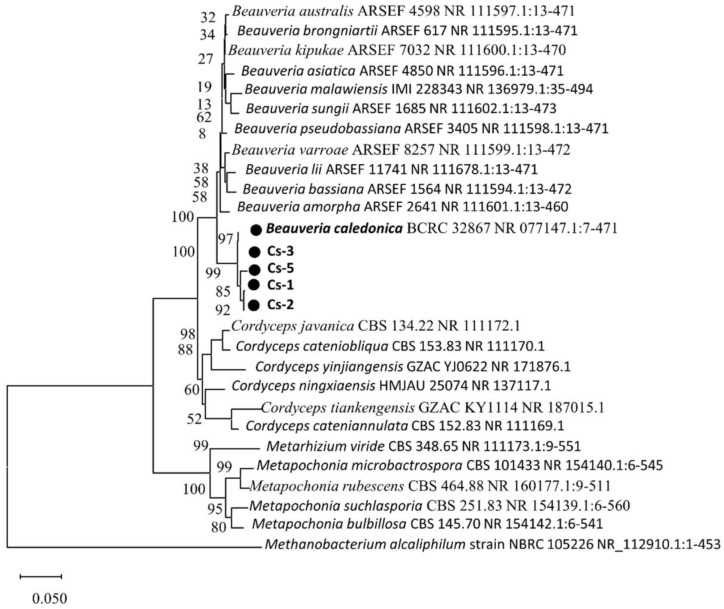
Phylogenetic tree of four *B. caledonica* strains based on rDNA-ITS sequence analysis.

**Figure 4 microorganisms-13-00782-f004:**
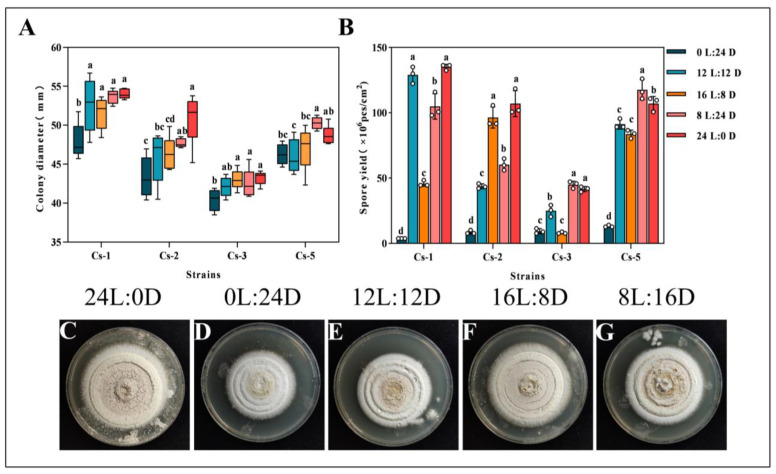
Effect of the photoperiod on the growth of the *B. caledonica* Cs-1 strain: (**A**) diameter of the four strains after 18 d of different light conditions; (**B**) spore production of the four strains after 18 d of different light conditions; (**C**–**G**) colony diagrams of the strains after 18 d of different light conditions. Different lowercase letters indicate significant differences at the 0.05 level, according to Duncan’s test. The white dots in the figure represent the actual measured values.

**Figure 5 microorganisms-13-00782-f005:**
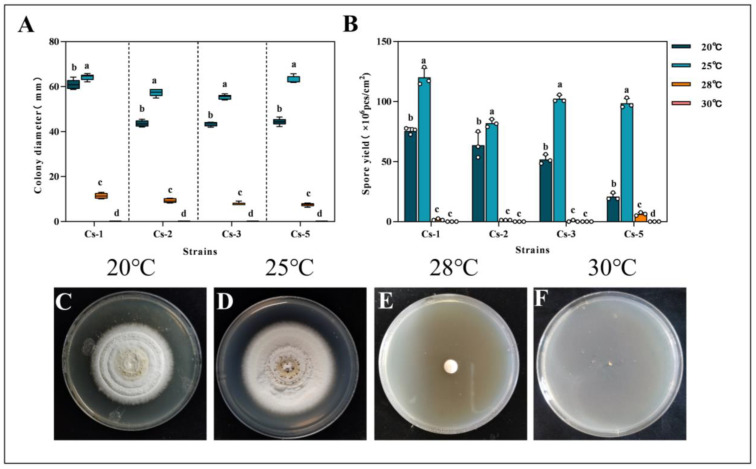
Effect of temperature on the growth of *B. caledonica*: (**A**) diameter of four strains after 18 d at different temperatures; (**B**) spore production of four strains after 18 d at different temperatures; (**C**–**F**) colony diagrams of strains after 18 d at different temperatures. Different lowercase letters indicate significant differences at the 0.05 level, according to Duncan’s test. The white dots in the figure represent the actual measured values.

**Figure 6 microorganisms-13-00782-f006:**
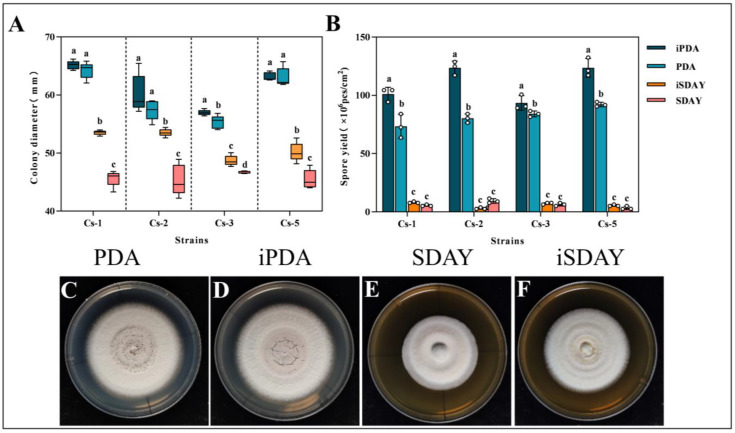
Effect of medium on the growth of *B. caledonica*: (**A**) diameter of 4 strains after 18 d under different medium conditions; (**B**) spore production of 4 strains after 18 d under different medium conditions; (**C**–**F**) colony diagrams of the strains after 18 d under different medium conditions. Different lowercase letters indicate significant differences at the 0.05 level, according to Duncan’s test. The white dots in the figure represent the actual measured values.

**Figure 7 microorganisms-13-00782-f007:**
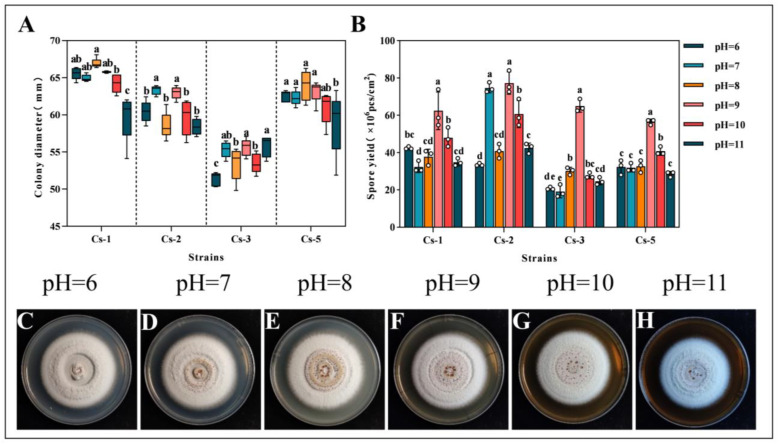
Effect of pH on the growth of *B. caledonica*: (**A**) diameter of the four strains after 18 d at different pH conditions; (**B**) spore production of the four strains after 18 d at different pH conditions; (**C**–**H**) colony diagrams of the strains after 18 d at different pH conditions. Different lowercase letters indicate significant differences at the 0.05 level, according to Duncan’s test. The white dots in the figure represent the actual measured values.

**Figure 8 microorganisms-13-00782-f008:**
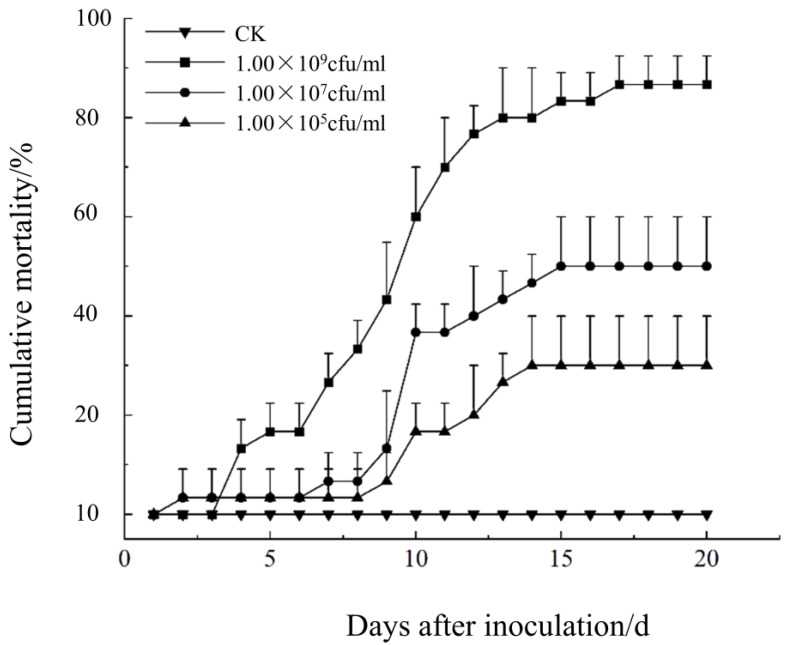
Comparison of different concentrations of spore suspensions on the cumulative mortality of adult *O. longicollis*.

**Figure 9 microorganisms-13-00782-f009:**
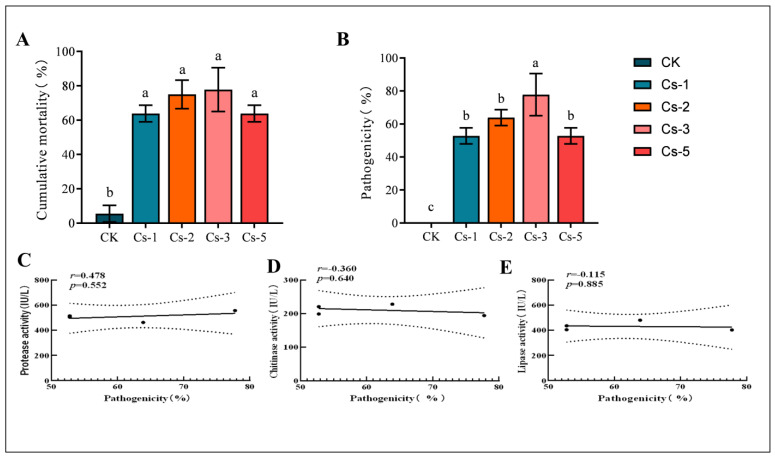
Enzyme activity and virulence of 4 strains of *B. caledonica*: (**A**) graph of cumulative mortality of the 4 strains; (**B**) graph of virulence of the 4 strains; (**C**) correlation of protease with virulence; (**D**) correlation of chitinase with virulence; (**E**) correlation of lipase with virulence. Different lowercase letters indicate significant differences at the 0.05 level, according to Duncan’s test. (**C**–**E**) Each point represents a set of data on pathogenicity and the corresponding enzyme activity. The lines represent the fitting relationships between variables.

**Figure 10 microorganisms-13-00782-f010:**
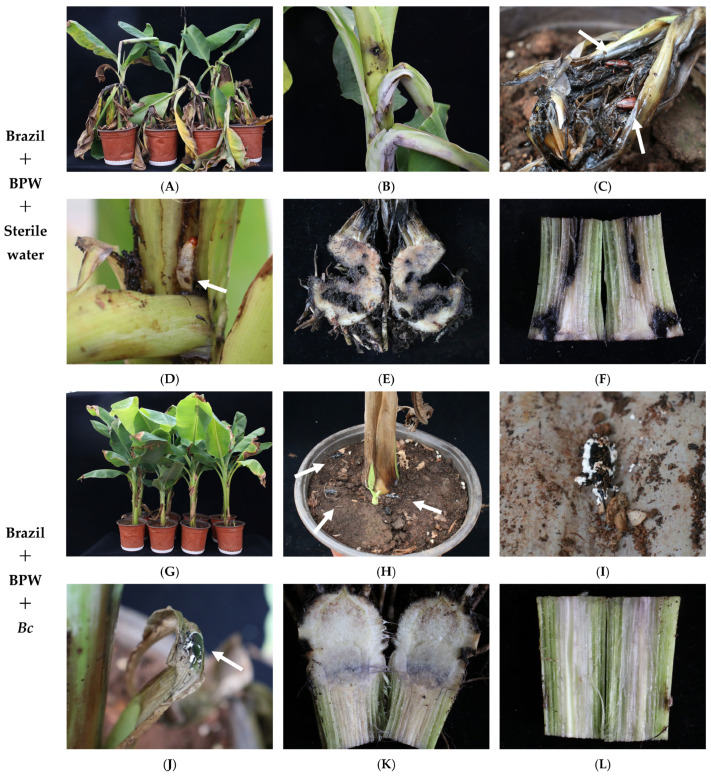
Interaction between *B. caledonica*, *O. longicollis* (Oliver), and banana plantlets: (**A**) Overall picture of banana plantlets in the CK group; (**B**) infestation symptoms in the CK group; (**C**) *O. longicollis* (Oliver) in the plant pseudostem in the CK group. The arrow indicates the adult of *O. longicollis* (Oliver); (**D**) larvae produced after 35 d in the CK group. The arrow indicates the larva of *O. longicollis* (Oliver); (**E**) infestation symptoms in the bulb of banana plantlets in the CK group; (**F**) infestation symptoms in the pseudostem of banana plantlets in the CK group; (**G**) overall picture of banana plantlets in the treatment group; (**H**) infestation symptoms in the treatment group. The arrow indicates the diseased insect; (**I**) *O. longicollis* (Oliver) onset in the soil to form stipe in the treatment group; (**J**) *O. longicollis* (Oliver) onset on plant to form stipe in the treatment group. The arrow indicates the diseased insect; (**K**) banana plantlet bulb in the treatment group; (**L**) banana plantlet pseudostem in the treatment group.

**Figure 11 microorganisms-13-00782-f011:**
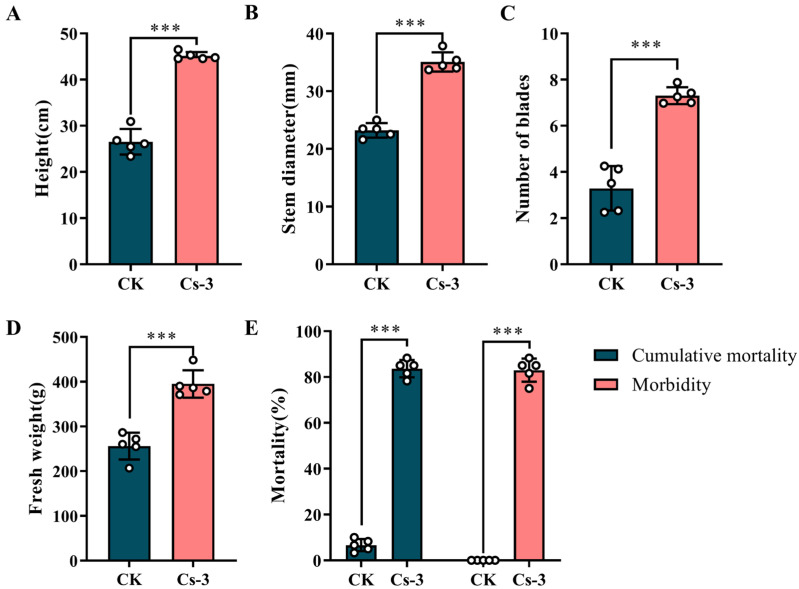
Physiological indices of banana plantlets under greenhouse conditions and pathogenicity of *B. caledonica*: (**A**) plant height of banana plants in different treatments; (**B**) stem thickness of banana plants in different treatments; (**C**) number of leaves of banana plants in different treatments; (**D**) fresh weight of banana plants in different treatments; (**E**) pathogenicity of *B. caledonica* against *O. longicollis* (Oliver). *** indicates a significant difference at the 0.001 level (*p*< 0.001) according to Duncan’s multiple-range test. The white dots in the figure represent the actual measured values.

**Table 1 microorganisms-13-00782-t001:** Pathogenicity of Cs-1 strains under different concentrations of spores against adult BPWs.

Conidia Concentration (cfu/mL)	Cumulative Mortality (%)	Regression Equation of Virulence	LT_50_/d	95% Confidence Limit
CK	0.00 ^c^	-	-	-
1.00 × 10^5^	20.00 ^bc^	-	-	-
1.00 × 10^7^	50.00 ^b^	y = −3.537 + 2.875x	17.00	15.254–19.583
1.00 × 10^9^	86.67 ^a^	y = −4.293 + 4.420x	9.36	8.713–9.998

Different lowercase letters in the same column indicate significant differences (*p* < 0.05). In the regression equation, y is the probability value, x is the natural logarithm of time, and “-” indicates that the final mortality rate of the adult BPW is less than 50%, and the LT_50_ cannot be estimated.

## Data Availability

The authors declare that all other data supporting the findings of this study are available in the paper and Appendix A or from the corresponding author(s) upon request. The raw data reported in this manuscript have been deposited in the Yunnan Key Laboratory of Green and Control of Agricultural Transboundary Pests, Agricultural Environment and Resources Institute, and the Yunnan Academy of Agricultural Sciences, Kunming.

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
