# Peer review of "Isolation and Characterization of Beauveria caledonica (Ascomycota: Hypocreales) Strains for Biological Control of Odoiporus longicollis Oliver (Coleoptera: Curculionidae)"

_microorganisms, 2025, doi:10.3390/microorganisms13040782_

Round 1

Reviewer 1 Report

Comments and Suggestions for Authors

The manuscript undoubtedly represents an exhaustive and detailed work. In addition to the above, the methodology has been carefully developed and adequately presented in the text, which is reflected in the results. However, in addition to some observations that I submit, I suggest a slight revision of the language in order to improve the work.

Reviewer 2 Report

Comments and Suggestions for Authors

Why did the author use a scanning electron microscope to observe mycelial morphology? The sample preparation for scanning electron microscopes involves a drying process, which could change the original morphology. Could the author provide morphological observation under an optical microscope?

Reviewer 3 Report

Comments and Suggestions for Authors

The article contains important information that can be used in practice. The manuscript is interesting, but I would like to suggest some minor changes.

The chapter 'Introduction" contains infromation about banana industry and pests that damage crops. I think that some detailed information about Odoiporus longicollis should be added. Information about morphology and life cycle is needed.

Research methods have been presented sufficiently. The charts are clear and most of the results are presented in detail and clearly, but there are concerns about the photos, which do not show clearly the symptoms of the pest's feeding. The quality of the photos should be improved so that the results of the observations can be seen.

Systematic affinity of all mentioned species should be complemented  both in the title and in the text.

Reviewer 4 Report

Comments and Suggestions for Authors

In this paper the authors isolate four strains of Beauveria caledonica and evaluate their potential as insect control treatments for banana plantations.  The authors find that the identified strains are pathogenic to the major banana plague Odoiporus longicollis.  

Major comments: 

  1. I recommend changing the title to "Isolation and Characterization of Beauveria caledonica strains for Biological Control of Odoiporus longicollis (Oliver) 

  1. The abstract must be rewritten, following the structure of scientific abstracts (i.e., background, methods, results and conclusions). 

  1. How was the Fungal DNA extracted (line 148)? 

  1. The description of the PCR method is incomplete. The concentrations of primers and template are missing (line 151 and 152). Which MasterMix was used? Commercial? 

  1. Which strains are shown in Fig.5 C-F, Fig 6 C-F and Fig 7C-H? 

  1. Which strain were used in the pathogenicity test (Fig. 8)? 

  1. What do the a and b mean on Fig. 9A and Fig 9B? 

  1. The method to inoculate the fungi into the insect is very different from what it could be achievable in a real-world application. How realistic is it to achieve significant insect infection in a real-world scenario. I think it would be interesting to discuss this point. 

  1. Can insects infect each other with the fungi? It might be interesting discuss this point in the discussion. 

  1. The authors identify the fungi in BPW from a banana plantation. Was that banana plantation performing better than other non-Bc ones?

Round 2

Reviewer 4 Report

Comments and Suggestions for Authors

The authors have addressed satisfactorily all my concerns.